# Kraft Lignin Grafted with Polyvinylpyrrolidone as a Novel Microbial Carrier in Biogas Production

**Agnieszka A. Pilarska [1],\*** , **Agnieszka Wolna-Maruwka [2]** and **Krzysztof Pilarski [3]**

1    Institute of Food Technology of Plant Origin, Poznań University of Life Sciences, ul. Wojska Polskiego 31, 60-637 Poznań, Poland
2    Department of General and Environmental Microbiology, Poznań University of Life Sciences, ul. Wojska Polskiego 31, 60-637 Poznań, Poland; amaruwka@up.poznan.pl
3    Institute of Biosystems Engineering, Poznań University of Life Sciences, ul. Wojska Polskiego 50, 60-637 Poznań, Poland; pilarski@up.poznan.pl
\*    Correspondence: pilarska@up.poznan.pl; Tel.: +48-61-848-73-08

**Abstract:** The objective of this study was to verify the effect of kraft lignin as a microbial carrier on biogas/methane yield. An anaerobic co-digestion test process was carried out, in which confectionery waste was used with sewage sludge. At the first stage of the study pure lignin and lignin combined with polyvinylpyrrolidone (PVP) were subjected to an extensive physicochemical analysis. Their morphology, dispersion and adsorption properties were determined. The two materials were also subjected to thermal, spectroscopic and elementary analysis. The anaerobic digestion of the two substrates was carried out with and without the addition of the carrier, under mesophilic conditions and in periodic operation. The monitoring and analysis of the two essential parameters, i.e., pH and volatile fatty acids/total alkalinity (VFA/TA) ratio, revealed that the process was stable in both tests. Microbial and biochemical analyses showed intensified proliferation of eubacteria and increased dehydrogenase activity in samples prepared with the lignin + PVP material. The cell count increased by 46% in the stuffed wafers (WAF) + sewage sludge (SS) variant with the carrier, whereas the enzyme activity increased by 43%. Cell immobilisation noticeably improved the process efficiency. The biogas production increased from 722 m$^3$ Mg$^{-1}$ VS to 850 m$^3$ Mg$^{-1}$ VS (VS – volatile solids), whereas the methane production increased from 428 m$^3$ Mg$^{-1}$ VS to 503 m$^3$ Mg$^{-1}$ VS (by about 18%). The research proved that lignin could be used as a very effective microbial carrier in anaerobic digestion (AD).

**Keywords:** anaerobic co-digestion; waste wafers; sewage sludge; carrier materials; process efficiency

## 1. Introduction

Microbial cell immobilisation in a suitable matrix ensures the stability of biochemical transformations, intensifies them and guarantees cost-effectiveness [1]. This method is based on microorganisms' natural capacity to adhere to surfaces, usually due to a deficit of nutrients in the environment. The concentration of nutrients is usually slightly higher near the surface. Therefore, cells adhering to surfaces have a metabolic advantage over cells freely suspended in a solution [2]. The choice of the right carrier is an important determinant of the success of the biotechnological process. The effective support material is selected by weighing various characteristics and required features of cell application against the properties/limitations/characteristics of combined immobilisation/support [3].

Principally, the microbe-material interactions lead to the formation of a biofilm. It causes degradation of the substrate in the anaerobic digestion process. Therefore, the choice of a high-efficiency

biofilm carrier is very important to enhance high-density methanogens. This may prevent the biomass from being washed out in the effluent. Earlier studies showed that the retention of microorganisms in anaerobic digestion (AD) reactors by biofilm carriers may potentially improve their productivity by increasing the number of methanogens and/or other mix cultures [4].

The application of microbial electrosynthesis (MES) for chemical synthesis from carbon dioxide ($CO_2$) is a very interesting solution, which improves biofilm growth. MES provides a novel approach to waste treatment, carbon dioxide fixation and renewable energy storage as it is based on a chemically stable compound, such as methane [5]. The key components that affect the functioning of the method are: Microbial catalysts, electrode materials, and the reactor design. Experiments conducted on different materials suggest that three-dimensional cathodes, which can retain more biomass, especially in hydrogen-based bioconversions, provide opportunities for further improvements in production efficiencies [6].

So far, cell immobilisation has rarely been used in anaerobic digestion (AD). Nevertheless, according to the state of the art, the use of a microbial carrier in the digestion process may result in higher methane yield and/or better gas quality. The immobilisation of microorganisms on various zeolites was also found positive, especially when clinoptilolite was used [7]. A granulated polymeric support [poly(acrylonitrile-acrylamide)] [4] and rubberised coir [8] were also tested as support media in the AD process. In search of adequate support matrices for efficient carbon cycling in waste regeneration, researchers investigated various materials, including clays, glass beads, microcarriers and membranes [9]. Some researchers also attempted to improve biohydrogen production by immobilising bacterial cells with magnetite nanoparticles [10], granulated activated carbon, wood shavings and perlite [11].

In search of natural and effective microbial carriers in the AD process, the authors of this study tested the suitability of kraft lignin. It is the second most abundant biopolymer after cellulose, and it is a by-product of the paper and pulp industry. Lignin has various advantages as a potential microbial carrier in biogas production in the AD process. It is characterised by low cost, good availability, non-toxicity, porosity, relatively good thermal stability and biocompatibility [12]. Apart from that, lignin is not digested by hydrolytic enzymes, so its degradation in the AD process is practically impossible [13]. So far there have been few reports describing the use of lignin as a carrier in biotechnological processes [14]. Although none of these publications concerned the use of lignin in the AD process, the authors clearly stated that lignin was characterised by very good sorption and it could be successfully used as a carrier for the immobilisation of bacteria and specific enzymes [15]. The material also draws significant attention, due to the sorption of heavy metals from aqueous media [12]. This property may significantly affect the choice of lignin in the process of anaerobic production of biogas from sewage sludge.

The aim of this study was to assess the influence of kraft lignin grafted with polyvinylpyrrolidone as a microbial carrier on the yield of biogas/methane produced in anaerobic co-digestion of stuffed wafers (WAF) and raw sewage sludge (SS). As waste wafers have a high concentration of carbohydrates, they are a promising substrate for biogas production [16]. Pilarska (2018) [17] demonstrated in earlier studies that SS was an adequate co-substrate for waste WAF. Sewage sludge is characterised by high buffering capacity and high content of organic matter [18]. At the first stage of the experiment the morphological, dispersive and physicochemical properties of pure kraft lignin and kraft lignin grafted with PVP were compared. Next, the biogas production efficiency was tested in a laboratory, in anaerobic batch reactors, under mesophilic conditions.

## 2. Materials and Methods

### 2.1. Materials

Stuffed wafers (WAF) and raw sewage sludge (SS) were the waste materials used in the research. The wafers were acquired from a factory in Poznań, whereas the sludge came from the Aquanet S.A.

sewage treatment plant in Poznań. Digested sewage sludge, which was used as the inoculum in the experiment, came from the biogas plant situated at the same sewage treatment plant. Kraft lignin was used as a microbial carrier, whereas polyvinylpyrrolidone (PVP)—a synthetic and non-toxic polymer was used as a modifier increasing cell adhesion. Both compounds were purchased from Sigma-Aldrich, Germany. Polyvinylpyrrolidone has excellent wetting properties and it readily forms films. Apart from frequent applications in the pharmaceutical and cosmetic industries, PVP is also used to improve bacterial adhesion [19].

### 2.2. Experimental Batch System Setup

#### 2.2.1. Substrate/Inoculum Batch Preparation

At the beginning of the experiment batches for the co-digestion of stuffed wafers with raw sewage sludge with (WAF + SS + Lignin) and without the carrier (WAF + SS) were prepared. There was also a sample with the inoculum and another one with lignin (Inoculum + Lignin) to verify and measure the amount of biogas produced from these materials. The substrate/inoculum ratios in the batches were calculated according to VDI Guideline 4630 (2006) [20] concerning the digestion of organic materials, characterisation of substrates, sample taking, collection of material data and digestion tests. It lists the conditions that must be met to check the biogas efficiency of substrates properly and characterises the type of inoculum that should be used. The inoculum should come from a sewage sludge digester or from a biogas plant with a similar profile of biogasified materials to the substrate tested. It should contain 1.5–2% of organic dry matter and there should be less than 10% of total solids (TS) in the batches to guarantee adequate mass transfers. The pH values of the batches ranged from 6.8 to 7.3. Although methanogens can tolerate a wide pH range, the neutral pH is the most favourable for their best metabolism and growth [21].

Table 1 shows the amount of the substrate and inoculum in the batches and their basic physicochemical properties.

**Table 1.** Composition and selected properties of the samples.

| Sample | Substrates (g) | Inoculum (g) | Lignin + PVP (g) | pH | Cond. (mS cm$^{-1}$) | TS (%) | VS (%) |
|---|---|---|---|---|---|---|---|
| WAF + SS/Inoc. | 133.2 [a] | 1070.0 | - | 7.11 | 77.97 | 3.76 | 68.50 |
| WAF + SS + Lignin/Inoc. | 133.8 [b] | 1069.0 | 24.0 | 7.15 | 78.85 | 3.74 | 69.78 |
| Inoculum | - | 1200.0 | - | 7.22 | 85.24 | 3.17 | 69.82 |
| Inoculum + Lignin | - | 1200.0 | 24.0 | 7.13 | 83.88 | 4.85 | 71.01 |

[a] 133.24 = 6.8(WAF) + 126.4(SS); [b] 133.24 = 6.8(WAF) + 127.0(SS). WAF, stuffed wafers; PVP, polyvinylpyrrolidone; TS, total solids.

#### 2.2.2. Bacterial Cell Immobilisation

The lignin + PVP carrier was prepared by wet mechanical mixing of 20 g of lignin ($C_9H_{10}O_2$, $C_{10}H_{12}O_3$, $C_{11}H_{14}O_4$) and 4 g of polyvinylpyrrolidone ($-[CH_2CH(C_4H_6NO)]_n-$), which were applied to particular substrate/inoculum batches (see Table 1) and stirred vigorously. The quantities of carrier components used in the experiment corresponded to those provided in reference publications [11,19].

#### 2.2.3. Biogas Production and Analysis

The rates of biogas production and biogas and methane yields were analysed according to the German standard DIN 38 414-S8 (1985) [22]. Each substrate and the control sample (inoculum) were digested in triplicate. Adequate substrate mixtures were placed in 1.4 L biofermenters (5) with 1.2 L of the feed in each (Figure 1). The material was stirred every 24 h to prevent any uncontrollable decay of the organic matter and to ensure the effectiveness of the carrier. Each biofermenter was equipped with a water jacket (4) connected to a heater (1). This enabled temperature control. The tests were carried

out under mesophilic conditions (at approx. 39 °C). The resulting biogas was transported through tubes (7) into tanks filled with a neutral liquid (8). In accordance with VDI Guideline 4630 (2006) [20], the experiment was conducted for each substrate until the daily biogas production was lower than 1% of the total amount generated.

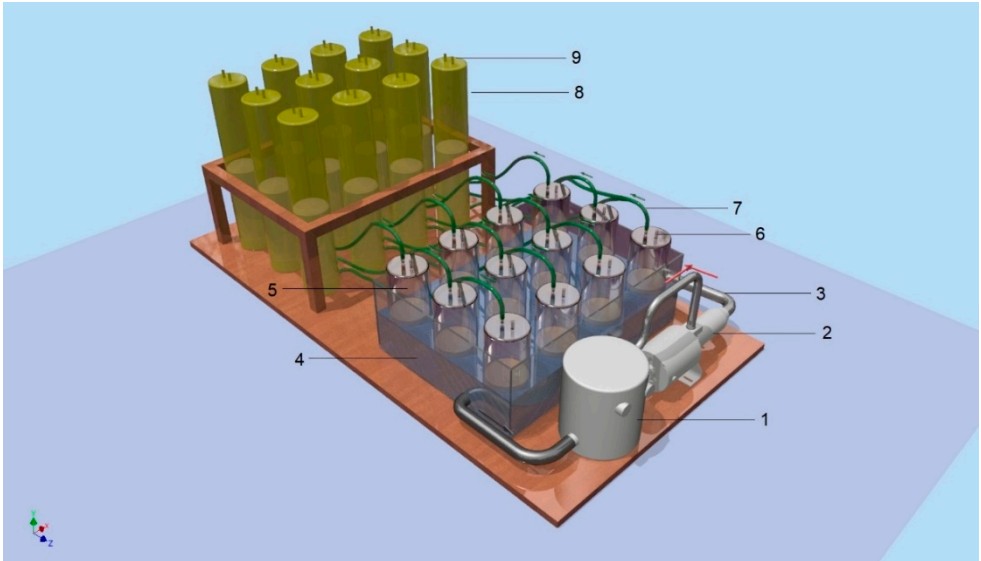

**Figure 1.** The biofermenter used in biogas production tests (18-chamber section); 1—water heater with temperature adjustment; 2—water pump; 3—insulated tubes for liquid heating medium; 4—water jacket (39 °C); 5—biofermenter (1.4 L); 6—slurry-sample drawing tube; 7—tube for biogas transport; 8—graduated tank for biogas; 9—gas sampling valve.

The generated gas volumes were measured every 24 h. The gas volumes of at least 1 L were analysed qualitatively—initially once a day, then—as lower volumes were generated—every three days. The biogas composition was measured with a GA5000 multifunctional portable gas analyser (Geotech, Leamington Spa, UK), which is adapted to measurements of $CH_4$, $CO_2$, $O_2$, $H_2$, and $H_2S$ at the following accuracies: $CH_4$ (0–70 vol % ± 0.5%), $CH_4$ (70–100 vol % ± 1.5%), $CO_2$ (0–60 vol % ± 0.5%), $CO_2$ (60–100 vol % ± 1.5%), $O_2$ (0–25 vol % ± 1.0%), $H_2S$ (0–5000 ppm ± 2.0%), and $H_2S$ (0–10,000 ppm ± 5.0%). A dual-wavelength infrared sensor was used to measure media ($CH_4/CO_2$) and an internal electrochemical sensor was used to measure $O_2/H_2S$. The measurements were taken upstream of the $H_2S$ filter to eliminate inaccuracies. The gas monitoring system was calibrated once a week by means of calibrating mixtures from Air Products [23].

### 2.2.4. Cumulative Biogas and Methane Estimate

After the qualitative and quantitative analyses of the gas, the final step was to assess the biogas yield per unit ($m^3$ $Mg^{-1}$) of organic dry matter. The biogas yield from the WAF + SS and WAF + SS + Lignin samples was calculated by subtracting the volume of gas generated from the Inoculum and from the Inoculum + Lignin. The ratio of gas generated from the inoculum in the batches in the reactors filled with the substrate mixtures was calculated according to the following equation:

$$V_{IS(corr.)} = \frac{\Sigma V_{IS} m_{IS}}{m_M},$$ (1)

where:
$V_{IS(corr.)}$ is the volume of gas released from the inoculum ($mL_N$); $\Sigma V_{IS}$ is the total volume of gas released from the inoculum during the test ($mL_N$); $m_{IS}$ is the mass of the inoculum used for the mixture (g); and $m_M$ is the mass of the inoculum used in the control test (g).

The specific volume of digestion gas $V_S$ produced from the substrate during the test was calculated step by step (from reading to reading) according to the following equation:

$$V_S = \frac{\Sigma V_n 10^4}{m w_T w_V},$$

(2)

where:

$V_S$ is the specific volume of digestion gas produced relative to the mass loss on ignition during the test ($L_N$ kgGV$^{-1}$); $\Sigma V_n$ is the net volume of gas produced from the substrate during the test (mL$_N$); $m$ is the mass of the weighed-in substrate (g); $w_T$ is the dry residue of the sample (%); and $w_V$ is the loss on ignition (GV) of the dry mass of the sample (%).

*2.3. Analytical Methods*

2.3.1. Physicochemical Characteristics of Substrates, Samples and Digestate

The substrates and batches were subjected to pH (potentiometric analysis) and electrolytic conductivity measurements (conductivity analysis) by means of an Elmetron CP-215 apparatus. The same materials were subjected to measurements of total solids (TS) by drying at 105 °C (Zalmed SML dryer) and volatile solids (VS) by burning at 550 °C (MS Spectrum PAF 110/6 furnace)—gravimetric analysis.

The following methods were used to measure the content of substrates:

- carbon—combustion at 900 °C,
- $CO_2$—infrared spectrometry, OI Analytical analyser,
- nitrogen—titration, Kjeldahl method with 0.1 n HCl and Tashiro's indicator,
- ammonium nitrogen—distillation and titration with 0.1 n HCl and Tashiro's indicator,
- phosphorus—mineralisation of phosphorus compounds with nitric acid in a microwave furnace (Milestone) and spectrophotometric analysis (Varian Cary 50).

The materials were also analysed for:

- the chemical oxygen demand (COD) by means of titration, dichromate method (potassium dichromate, concentrated sulfuric acid, silver sulphate as a catalyst),
- the content of light metal ions by means of inductively coupled plasma optical emission spectrometry (ICP-OES), using a JY 2000 2 ICP-OES spectrometer,
- the content of sewage sludge and heavy metal inoculum by means of atomic absorption spectrometry (AAS) after dry mineralisation with a Hitachi Z-8200 atomic absorption spectrometer.

5 mL of a particular sample (5 mL of the fermentation substrate) was titrated with 0.1 N of a sulfuric acid solution ($H_2SO_4$) up to pH 5.0 to measure the content of volatile fatty acids (VFA), total alkalinity (TA) and the ratio between volatile fatty acids and total alkalinity (VFA/TA ratio) in the digestate.

The analyses of these parameters were triplicated.

2.3.2. Physicochemical Characteristics of Pure Lignin and Lignin + PVP System

The research included analyses that were necessary for comparative assessment of the morphological, dispersive and adsorptive properties of pure lignin and lignin grafted with PVP.

Fourier transform infrared (FTIR) spectroscopy measurements were made with a Vertex 70 spectrophotometer (Bruker, Germany) at room temperature. The samples under analysis had the form of tablets, made by pressing a mixture of anhydrous KBr (ca. 0.25 g) and 1 mg of the tested substance in a special steel ring under a pressure of approximately 10 MPa. FTIR spectra were obtained in the transmission mode between 4000 and 400 cm$^{-1}$. The analysis was made at a resolution of 0.5 cm$^{-1}$.

The chemical composition of the samples was identified with a FLASH 2000 elementary analyser (Thermo Fisher Scientific, Waltham, MA, USA), which applied the dynamic combustion method. Samples (about 2–4 mg) were placed in the reactor by means of an autosampler with a strictly specified portion of oxygen. After combustion at 900–1000 °C, the exhaust gases were transported in the helium flow to the other furnace of the reactor filled with copper. Next, they came through the water trap to the chromatographic column, which separated individual products. Finally, the separated gases were detected by a thermal conductivity detector.

A Zetasizer apparatus equipped with a 4 mW helium/neon laser (Zetasizer Nano ZS, Malvern Instruments Ltd., Malvern, UK) was used to measure the size of particles. It can measure particles sized 0.6–6000 nm (non-invasive backscattering technique—NIBS). Before the analysis all samples were dispersed in isopropanol by means of mild sonication. All the measurements were taken at 25 °C.

The cumulant analysis gives a width parameter known as the polydispersity, or the polydispersity index (PdI). The cumulant analysis is actually the fit of a polynomial to the log of the *G*1 correlation function:

$$ln[G1] = a + bt + ct^2 + dt^3 + et^4 + \ldots \tag{3}$$

The value of *b* is known as the second-order cumulant, or the z-average diffusion coefficient. The coefficient of the squared term, *c*, when scaled as $2c/b^2$, is known as the polydispersity.

The morphology and microstructure of the lignin and lignin + PVP materials were examined in SEM (Scanning Electron Microscope) images recorded with an FEI Quanta FEG 250 microscope (Thermo Fisher Scientific, USA). The microscope worked in the Low Vacuum mode at 70 Pa and accelerating voltage of 10 kV. Before the test the samples were coated with gold for 5 s by means of a Balzers PV205P coater (Balzers, Switzerland).

Nitrogen adsorption/desorption isotherms at 77 K and porous structure parameters, such as the surface area ($A_{BET}$), total volume ($V_p$) and mean size ($S_p$) of pores were measured with an ASAP 2420 instrument (Micromeritics Instrument Co., Norcross, GA, USA). Before the samples were degassed in a vacuum at a temperature of 90 °C. The specific surface area (SSA) was calculated using the Brunauer–Emmett–Teller (BET) model applied to the linear part of the adsorption isotherm ($0.05 < P_0 < 0.25$). The BJH (Barrett–Joyner–Halenda) method was applied to measure the mean pore size and volume.

The thermal stability of pure lignin and the PVP + lignin material was analysed by means of a TGA4000 instrument (Perkin Elmer, Waltham, MA, USA). The tests were carried out in a nitrogen atmosphere. The samples were heated from 25 °C to 995 °C in nitrogen flow (20 mL min$^{-1}$). They were kept at 995 °C for 1 min and then they were cooled down.

2.3.3. Microbial Analysis of Digestate

The total bacterial count in the six samples was identified directly under a fluorescence microscope (Carl Zeiss) by means of fluorescent in situ hybridisation (FISH), which was modified according to Amann et al. (1990a) [24]. There were four time points of microbial analyses: The first at the beginning of the experiment; the second after 5 days; the third after 8 days; and the fourth after 14 days.

The digested material (0.01 mL) was placed on the surface of microscope slides by means of a Breed pipette and then it was fixed with a 4% paraformaldehyde (PFA) solution. At the next stage the samples were washed in a PBS solution three times and 0.5% Triton solution was added. Then the samples were washed in the PBS solution three times again. Next, they were placed in an alcohol series (70%, 80%, 96%). When 70% formamide solution was added, the genetic probe EUB338 GCT GCC TCC CGT AGG AGT was applied [25]. It was concentrated at 25 ng uL$^{-1}$, marked with Cy5 fluorescent dye and suspended in a solution consisting of 5 M NaCl, 1 M Tris/HCl, 25% formamide, 10% SDS and ddH$_2$O.

After 24-h incubation of the digestate samples in darkness at a temperature of 37 °C they were analysed by means of a Zeiss AxioImager M1 fluorescence microscope equipped with a

colour digital camera, AxioCam MRc5. The image was analysed with the AxioVision 4.8 software (Birkerød, Denmark).

At the last (5th) time point of analysis the aforementioned in situ hybridisation method was used to measure the count of methane microorganisms of the Archaea domain. In order to detect these microorganisms in the digested waste samples the ARCH915 GTG CTC CCC CGC CAA TTC CT probe marked with Cy5 fluorescent dye was applied, as suggested by Stahl and Amann (1991) [26].

Apart from that, the digestate sample with the largest count of microorganisms (eubacteria and Archaea) was subjected to analysis of bacterial colonisation on the surface of the carrier, i.e., lignin. The sample was analysed with a Hitachi SU3500 scanning electron microscope, which enables observation of samples magnified 5–100,000 times. The SEM was equipped with a BSE-3D detector (backscattered electron image). During measurements the pressure in the microscope chamber was 50 Pa. The electron beam size was 50, whereas its intensity ranged from 86,900 to 93,100 nA.

### 2.3.4. Biochemical Analysis of Digestate

Spectrophotometry was applied for biochemical analysis. The dehydrogenase activity (DHA) was determined according to the method developed by Thalmann (1968) [27] with some minor modifications. The waste (5 mL) was incubated for 24 h with 2, 3, 5-triphenyltetrazolium chloride (TTC) at 30 °C, pH 7.4. Triphenylformazan (TPF) was produced, extracted with 96% ethanol and measured spectrophotometrically at 485 nm. The dehydrogenase activity was expressed as $\mu$mol TPF $g^{-1}$ DM of waste 24 $h^{-1}$.

### 2.3.5. Statistical Analysis

The data were statistically processed with the Statistica 12.0 program (StatSoft Inc. 2012, Tulsa, OK, USA). Two-way analysis of variance was used to verify the significance of variations in the count and activity of the microorganisms under study according to the experimental variant and time of analysis.

## 3. Results and Discussion

### 3.1. Substrate and Inoculum Characterisation

The pH of both substrates and the inoculum was close to neutral (see Table 2). Sewage sludge (SS) had the lowest pH, i.e., 6.23. The inoculum, i.e., digested sewage sludge, was characterised by the highest conductivity (30.03 mS $cm^{-1}$), due to the content of inorganic ions, formed as a result of the biodegradation of substances in the raw sewage sludge. Waste wafers (WAF), which are a solid material, were characterised by high content of total solids (TS), i.e., 98.03 wt %, whereas sewage sludge, which is a suspension, had very low TS content, i.e., 5.06 wt % in raw SS and 2.70 wt % in the inoculum. Due to the low TS content in the sludge it is justified to carry out anaerobic co-digestion of this substrate with other solid organic materials, including food waste [28].

The substrates used in the experiment, including the inoculum, were characterised by high content of volatile solids (VS). The highest VS content was noted in the WAF (98.50 wt % TS), which is in agreement with the results provided in reference publications [16,17]. As shown in Table 2, the high content of organic matter in the wafers was accompanied by the high C/N ratio, i.e., 46.21, due to large amounts of carbohydrates and fats. According to some authors [29], the increased availability of carbon in the digester (due to the use of food waste as a co-substrate), not only makes it possible to balance the C/N ratio of the digester, but it also helps to avoid ammonia inhibition, which is often mentioned when SS is used. In general, the high C/N ratio prevailed in the WAF, but the SS was characterised by a low C/N ratio, i.e., 8.07 (see Table 2). This value was within the range indicated in reference publications [28]. Interestingly, the range can be increased to 6–15 by resultant mixing [28]. As reference publications show, the values of the parameters that might inhibit the AD process, including ammonium nitrogen and light metal ions, are safe for the process stability [21]. It is noteworthy that the chemical oxygen demand (COD) value of the inoculum was much lower

(1630 mg L$^{-1}$) than that of the SS (3200 mg L$^{-1}$). Significant differences in the value of this parameter between the digested and raw sludge are caused by the breakdown of inorganic compounds, including sulphites and sulphides, in the AD process. The COD values measured in the experiment corresponded to the results provided in reference publications [29]. Other studies published by the author of the experiment include data on the content of heavy metals in both sludge types [17,18].

**Table 2.** Physicochemical properties of substrates and inoculum used in the experiment.

| Waste | pH | Cond. | TS | VS | C/N Ratio | C | N | N-NH$_4$ | P | COD | VFA | Light Metal Ions | | | |
|---|---|---|---|---|---|---|---|---|---|---|---|---|---|---|---|
| | | | | | | | | | | | | K | Na | Mg | Ca |
| | - | (mS cm$^{-1}$) | (wt %) | (wt %$_{TS}$) | - | (wt %$_{TS}$) | (wt %$_{TS}$) | (wt %$_{TS}$) | (mg kg$^{-1}$) | (mg L$^{-1}$) | (mg L$^{-1}$) | (mg kg$^{-1}$) | | | |
| Waste wafers (WAF) | 6.80 | 1.94 | 98.03 | 98.50 | 46.21 | 41.59 | 0.90 | 0.29 | 152 | 1420 | ND | 37 | 154 | 49 | 154 |
| Sewage sludge (SS) | 6.23 | 3.47 | 5.06 | 81.38 | 8.07 | 38.40 | 4.76 | 0.37 | 1234 | 3403 | 3500 | 1900 | 3671 | 17 | 36 |
| Inoculum | 7.19 | 30.03 | 2.70 | 67.64 | 3.33 | 27.68 | 8.32 | 3.86 | 2560 | 1630 | 360 | 3236 | 6300 | 33 | 52 |

ND, not determined; COD, chemical oxygen demand; VFA, volatile fatty acids.

### 3.2. Physicochemical Properties of Pure Lignin and Lignin + PVP System

#### 3.2.1. FTIR Spectroscopy

Both samples were subjected to FTIR analysis to confirm the nature of the combination of polyvinylpyrrolidone and lignin. The shape of the oscillation spectra shown in Figure 2 was almost identical. There were no bands that could correspond to functional groups other than those that were characteristic for pure lignin. Therefore, we can conclude that the combination of PVP and lignin was physical (mainly through hydrogen bridges) and it influenced only its surface properties rather than the chemical structure. It is noteworthy that the absence of a chemical reaction between these compounds was chiefly caused by their non-stoichiometric weight ratio (20 g of lignin: 4 g of PVP).

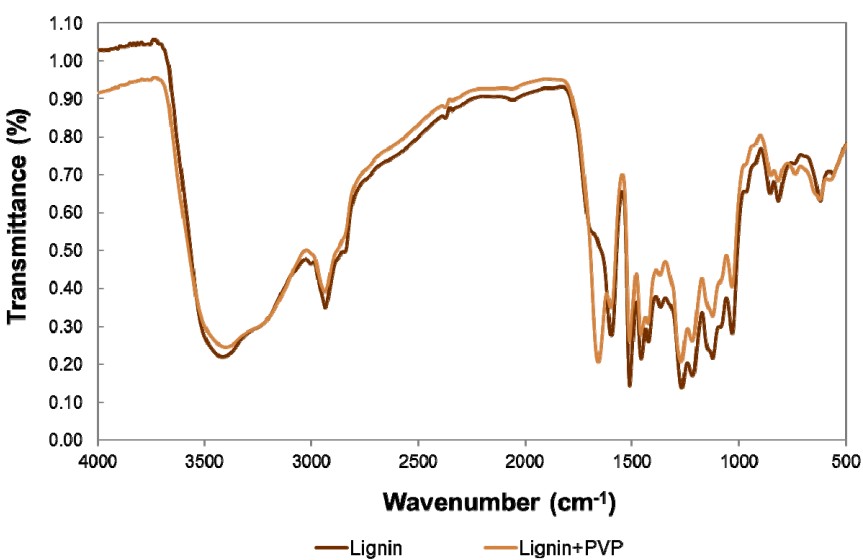

**Figure 2.** FTIR spectra of the lignin and lignin + PVP samples.

The spectrum of pure lignin (see Figure 2) shows bands assigned to the O–H stretching vibrations (3600–3200 cm$^{-1}$), C–H stretching vibrations (3004–2846 cm$^{-1}$), vibrations of the C=O ketone group (1710–1530 cm$^{-1}$) and the ones at 1599 cm$^{-1}$, 1512 cm$^{-1}$, 1426 cm$^{-1}$ and 1368 cm$^{-1}$ assigned to the stretching vibrations at the C–C bonds in the aromatic skeleton. Further analysis of the lignin spectrum shows another group of bands at 1270 cm$^{-1}$, 1218 cm$^{-1}$, 1125 cm$^{-1}$ and 1033 cm$^{-1}$ corresponding to the stretching vibrations of C–O and C–O–C ether bands. The spectrum shows a group of bands below 1000 cm$^{-1}$ assigned to the in-plane and out-of-plane vibrations of aromatic C–H bonds. This description

is in agreement with the data provided in reference publications [30]. The lignin + PVP spectrum is not significantly different from the pure lignin spectrum. Polyvinylpyrrolidone consists of analogous moieties, which give the same characteristic peaks in the spectrum as lignin. The combination of nitrogen (from the aromatic ring) with carbon did not have a clear band at 1200–1120 cm$^{-1}$.

### 3.2.2. Elemental Analysis

The elemental analysis of the pure lignin sample and the lignin + PVP sample also gave interesting results (see Table 3).

**Table 3.** Elemental content and parameters of porous structure properties of lignin and lignin + PVP samples.

| Materials | Elemental Content (%) | | | | Porous Structure Properties | | |
|---|---|---|---|---|---|---|---|
| | N | C | H | S | $A_{BET}$ (m$^2$/g) | $V_p$ (cm$^3$/g) | $S_p$ (nm) |
| Lignin | - | 60.29 | 5.61 | 0.57 | 1.9 | 0.01 | 18.4 |
| Lignin + PVP | 0.68 | 62.28 | 6.08 | 0.44 | 2.5 | 0.01 | 19.8 |

The significant content of carbon and hydrogen in lignin confirms its phenylpropanoid structure based on three fundamental units, namely p-coumaryl, sinapyl, and coniferyl alcohols [12]. The percentage of these components in the experiment was similar to the data provided in reference publications [30]. The presence of sulphur is an expected result of the way in which lignin is isolated in the kraft process. The content of approximately 0.57% matches the standards laid down for this biopolymer [30]. Interestingly, the lignin + PVP material contained less sulphur. It is an advantage when we consider the use of this material as a microbial carrier. The content of carbon and hydrogen in this sample was slightly higher than in pure lignin, due to the presence of polyvinylpyrrolidone. The content of nitrogen in the lignin + PVP sample resulted from the chemical composition of PVP, where the vinylpyrrolidone had the following formula: $-[CH_2CH(C_4H_6NO)]-$.

### 3.2.3. Dispersive and Morphological Properties

Another important part of the study was a comparative analysis of the dispersive and morphological properties of the lignin and lignin + PVP materials, as shown in Figure 3.

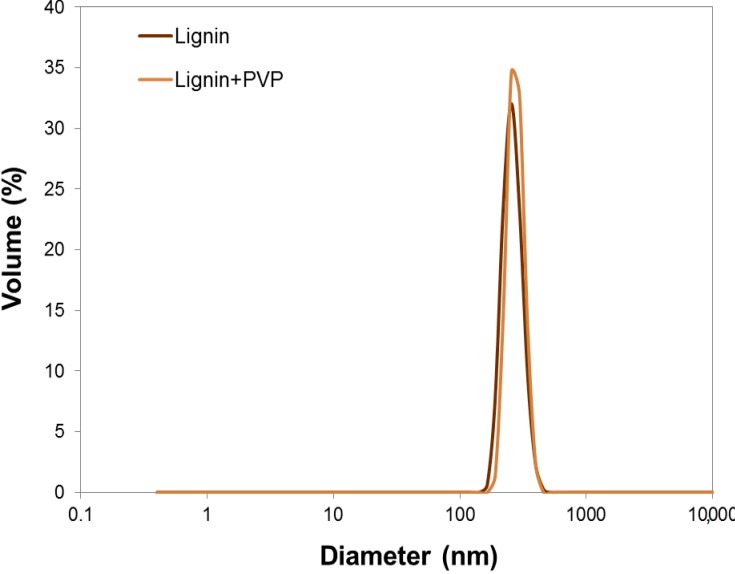

**Figure 3.** Particle size distributions by volume of the lignin and lignin + PVP samples.

The diagram shows individual, relatively narrow bands referring to both samples. The shape of the curves and the PdI coefficients of 0.225 and 0.342 for lignin and lignin + PVP, respectively, indicate the minimal distribution of particle sizes. The maximum share of lignin was 33% from particles sized 265 nm, whereas the maximum share of lignin + PVP was 35% from particles sized 280 nm. The results measured with a Zetasizer Nano ZS do not suggest changes in dispersion parameters resulting from the use of the PVP modifier. The particles in the lignin + PVP sample were slightly bigger. Both materials tended to form aggregate structures (up to 1 μm).

The enlargements of the SEM images of both samples (Figure 4a–d) show aggregates of irregularly shaped particles with a porous microstructure. According to reference publications, due to the extremely extensive phenylpropane structure of lignin it tends to form larger structures regardless of the additives used [30]. As can be seen in the images, the material with the PVP modifier had a slightly larger share of small particle aggregations than the pure lignin (Figure 4c,d). This fact may have positively affected the development of the specific surface of the lignin + PVP carrier (see the next subsection). It is noteworthy that recently Sameni et al. (2018) [31] applied the emulsion solvent evaporation (ESE) technique and obtained lignin microspheres of uniform spherical shape and narrow size distribution from a number of different lignins. They presented SEM images of the product, whose morphology was noticeably different from the kraft lignin morphology. In view of the increasing range of lignin applications, it is justified to develop preparation methods to create defined lignin properties.

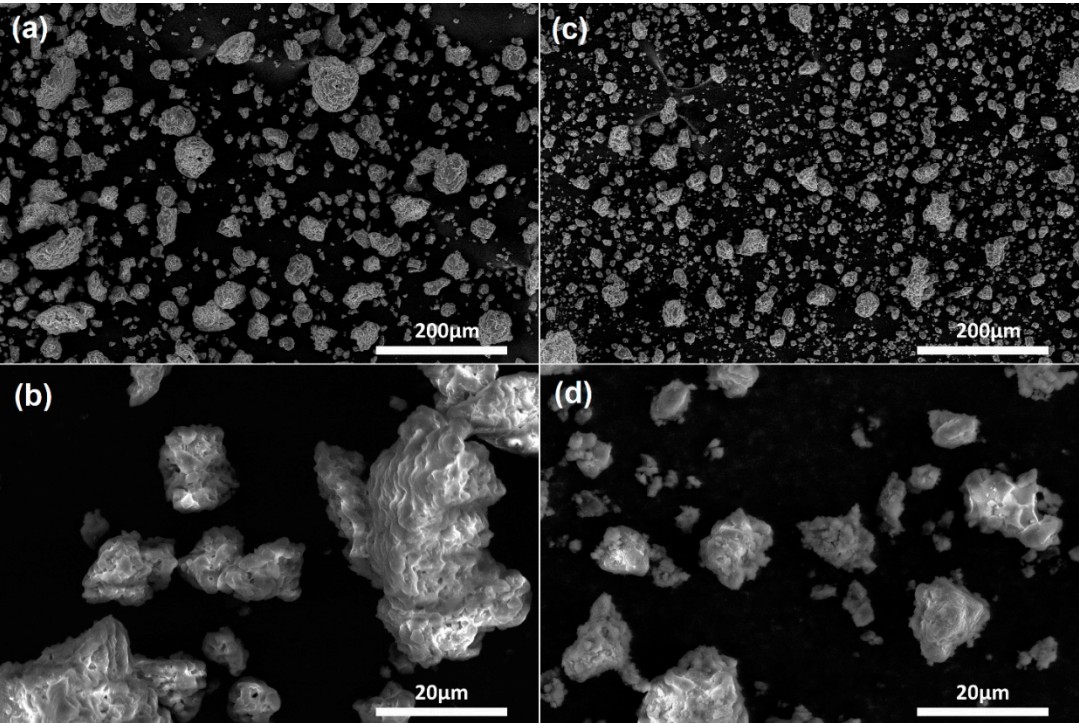

**Figure 4.** SEM images of (**a**,**b**) the lignin and (**c**,**d**) lignin + PVP samples at different magnifications.

### 3.2.4. Porous Structure Properties

The porous structure parameters of both samples were assessed to determine the possible applications of PVP-grafted lignin as a microbial carrier in the AD process and to obtain full information about the influence of the modifier on the physicochemical properties of pure lignin. Figure 5a shows the nitrogen adsorption/desorption isotherms of both samples. The character of the isotherms indicates their mesoporous structure. The quantity of nitrogen adsorbed by the samples increased slightly up to a relative pressure of 0.8. When the value of $p/p_0$ was exceeded, the quantity of nitrogen increased rapidly in both samples, although the quantity of adsorbed nitrogen was slightly greater in the lignin +

PVP sample. The BET surface area ($A_{BET}$), pore volume ($V_p$) and pore diameter ($S_p$) were calculated on the basis of the adsorption and desorption isotherms. Table 3 shows their values.

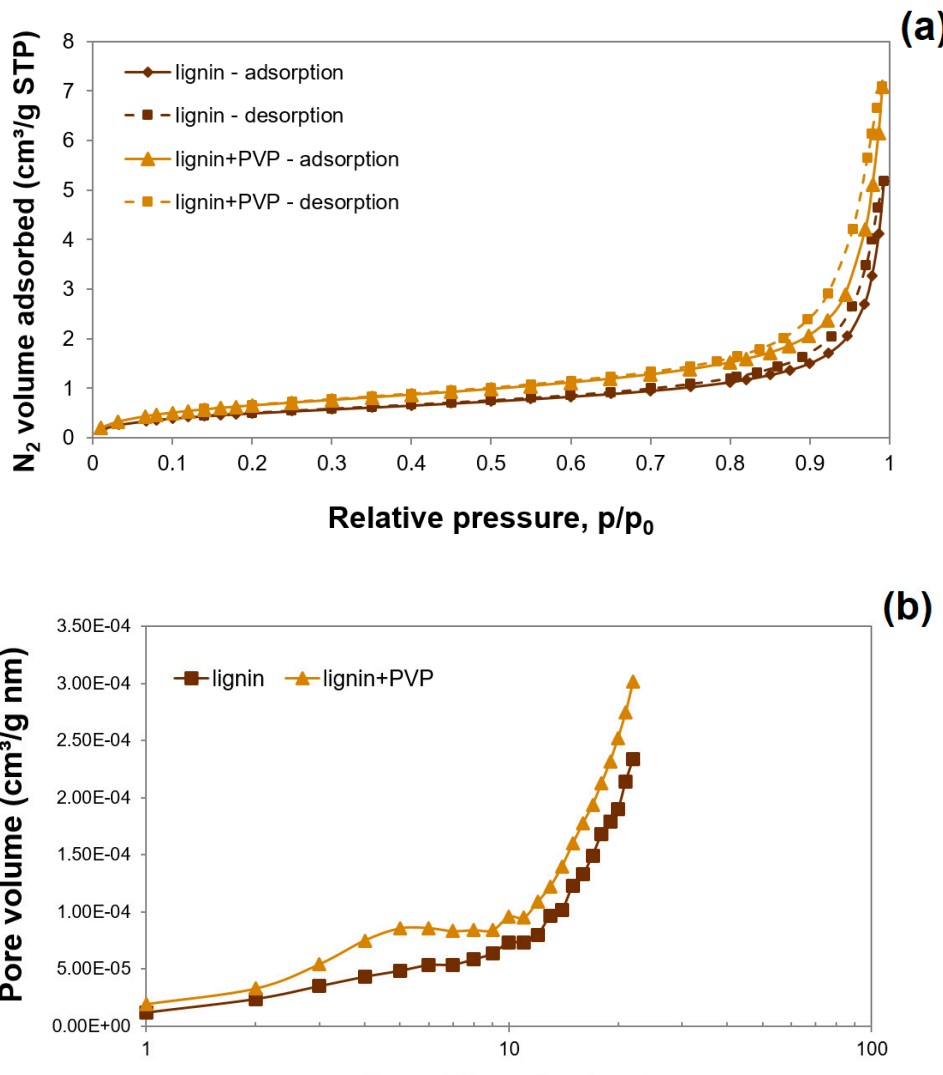

**Figure 5.** $N_2$ adsorption/desorption isotherms (**a**) and pore size distribution (**b**) of the lignin and lignin + PVP samples.

As the results in Table 3 show, the addition of PVP had a positive effect on the porous structure properties, but these were minor changes. However, it is noteworthy that this compound was primarily supposed to improve cell adhesion to the lignin surface. There have been no reports on PVP causing significant changes to the microstructural properties of materials. As the results show, pure lignin was characterised by relatively low BET surface area (1.9 m$^2$/g), whereas the BET surface area of lignin + PVP was only slightly greater, i.e., 2.5 m$^2$/g (see Table 3). The increase in the specific surface area of the material may have been caused by the increase in material porosity. The information about the noticeable mesoporous structure of this material was of key significance to this study. Both in pure and modified lignin the pore sizes were considerable (18.4 nm and 19.8 nm, respectively). These results are correlated with the SEM images shown above (Figure 4). Figure 5b shows that both materials had similar pore size distribution.

The lignin pore diameter was greater than the diameters of other organic and inorganic materials [32]. Thanks to this valuable property, lignin can be an excellent microbial carrier or sorbent. In order to make a lignin-based material with ideal porous structure properties for 'green' applications

researchers form hybrids by combining lignin with specific chemical compounds, which usually have a large specific surface area [30,33]. They also develop special methods of lignin synthesis [32]. The AD process with lignin increases the survival rate and activity of bacterial cells even under destabilised conditions. Large pores in lignin and the addition of PVP enable the effective formation of durable biofilm layers (see Section 3.3.2).

### 3.2.5. Thermal Analysis

Both samples were subjected to thermal gravimetric analysis (TGA) and differential thermal gravimetric (DTG) analysis to clarify thermal tolerance in the temperature–mass relation (as shown in Figure 6). It may be particularly important to know the temperatures of decomposition of lignin (or lignin combined with PVP) as a potential microbial carrier in anaerobic digestion when the process is carried out under thermophilic conditions (50–60 °C).

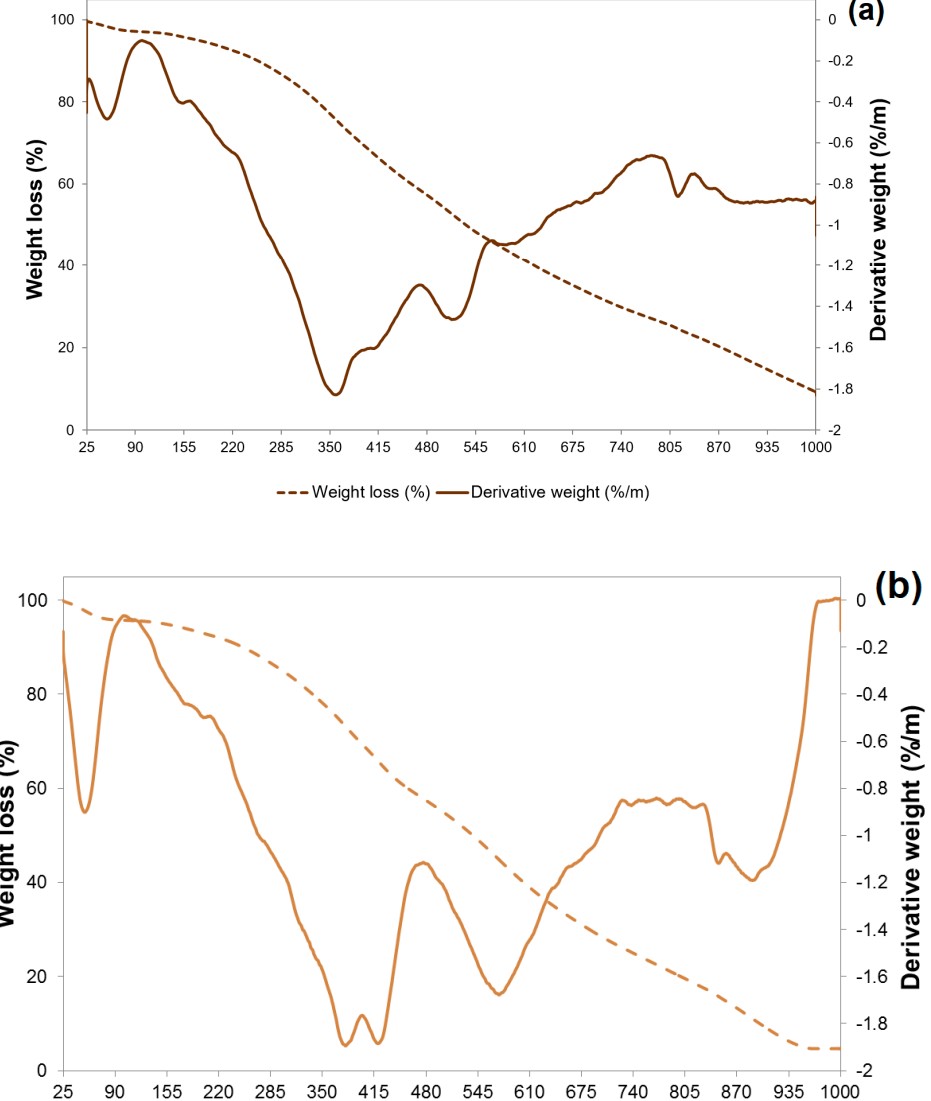

**Figure 6.** Thermal analysis thermal gravimetric analysis (TGA)/ differential thermal gravimetric (DTG) of (**a**) the lignin and (**b**) lignin + PVP samples.

The temperature-responsive mass evolution was a two-stage process both for pure lignin and the lignin + PVP sample, as TGA curves (dashed lines) in Figure 6a,b show. The first stage is accompanied by the loss of adsorbed/hydrated water within 31–71 °C in lignin and within 37–72 °C in lignin + PVP. According to reference publications, the loss of water in lignin takes place up to a temperature of about 100 °C [15], depending on the origin of lignin. According to Cui et al. (2013) [34], kraft lignin is susceptible to molecular mass changes at temperatures above ~ 120 °C. In fact, this information is sufficient to conclude that lignin (also when combined with PVP) is a suitable microbial carrier in the AD process thanks to its proven thermal stability. There is a minor weight loss below 200 °C. This conclusion was also confirmed by the results of the analysis in this study. The weight loss in lignin did not exceed 2.7% up to 214 °C, whereas in lignin + PVP it did not exceed 3.9% up to 192 °C (see DGT curve, solid lines). These results are very similar to the data provided by Moustaqim et al. (2018) [35] in their latest study on thermal analyses of lignin.

According to available sources of information [35], the main exothermal degradation stage takes place within 250–570 °C. This stage is divided into three substages associated with the release of various volatile compounds: Phenolic compounds with an aromatic ring, hydroxyl and alkyl groups (~270 °C), and the release of methanol and methane (starting at about 380 °C and 530 °C, respectively).

The thermal decomposition behaviour of lignin observed in this study was in agreement with the aforementioned data. The largest weight loss (33.9%) at the second stage of lignin pyrolysis occurred within 288–388 °C (Figure 6a), whereas the next one took place between 488 °C and 543 °C. The weight loss in the lignin + PVP sample was smaller. The largest weight loss (24.7%) occurred within a higher temperature range, i.e., 519–612 °C. The weight loss was smaller at lower temperature ranges, i.e., 15% within 341–394 °C and 11.5% within 401–444 °C. Thus, the second stage of thermal decomposition behaviour in pure lignin differed from lignin + PVP. The presence of PVP had a noticeable positive influence on the thermal stability of lignin. The increase in thermal stability of the modified sample was caused by the crosslinking of lignin and PVP. Reference publications confirm the fact that PVP is significant for the thermal stabilisation of other materials, including platinum nanoparticles [36]. Moustaqim et al. (2018) [35] also found differences in the thermomechanical behaviour of lignin and noted that they may have been caused by internal frictions characteristic of the thermally stressed material.

*3.3. Batch Digestion Test*

3.3.1. Process Stability

Most studies show that the system pH is the most important parameter, which greatly affects the AD process [37]. The initial pH of batches prepared in this experiment was neutral and ranged from 6.9 (WAF + SS + Lignin) to 7.3 (Inoculum), as can be seen in Figure 7 (dashed lines). Importantly, during the consecutive days of the process there were no significant decreases in the pH of the batches. The pH of the WAF + SS and WAF + SS + Lignin codigestion systems was increasing gradually until the 5th and 9th day, respectively. As the shapes of the curves show, after the 9th day the pH value of all the samples ranged from 7.35 to 7.6 and finally reached 7.5 (WAF + SS) and 7.55 (the other samples). The shortest retention time was noted in WAF + SS (17 days). The retention time of the other three trials was longer and identical (20 days). During the experiment there were no unfavourable pH drops in any of the systems. The anaerobic digestion of all the batches was stable. It proved again that waste wafers were an appropriate substrate for biogas plants, whereas sewage sludge perfectly buffered the system as its co-substrate [17]. According to reference publications, sewage sludge is enriched with various salts, so the proper SS ratio in food waste also buffers the system [38]. In anaerobic co-digestion systems alkalinity is a key factor stabilising pH variations resulting from the hydrolysis of highly biodegradable substrates, such as food waste, including confectionery waste [18].

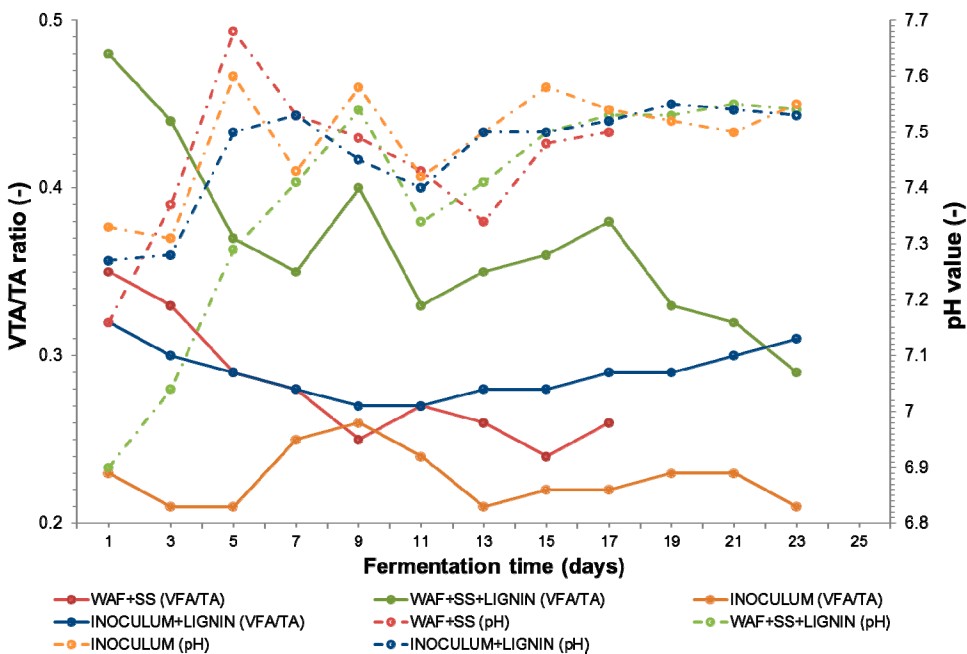

**Figure 7.** Variation in the pH and VFA/ total alkalinity (TA) ratio during the anaerobic digestion of the WAF+SS, WAF + SS + Lignin, Inoculum and Inoculum + Lignin samples.

The monitoring of the process stability included not only measurements of the pH value, but also the VFA/TA ratio. Having assumed the criteria of the VFA/TA ratio ranges, i.e., VFA/TA $\leq$ 0.40—stable digester, 0.40 < VFA/TA < 0.80—some instability, and VFA/TA $\geq$ 0.80—significant instability [39], we can say that the process was generally stable in all the batches. Only at the initial stage was the VFA/TA ratio measured in the combination of both co-substrates with lignin (WAF + SS + Lignin) was relatively high, i.e., 0.48. However, during the consecutive days of biodegradation of the substances in this batch this value was decreasing. It fluctuated within 'safe' limits to reach the final value of 0.29 (see Figure 7, green, solid curve). As regards the other samples, during the process the VFA/TA ratio values varied, but were well below the limit of 0.4. The decreasing value of this parameter indicates the gradual depletion of organic carbon compounds and confirms the high activity of bacteria. This observation is in agreement with other researchers' findings, who described the VFA/TA ratio variability during the anaerobic digestion of sewage sludge and other organic wastes [40].

3.3.2. Enzyme Activity and Counts of Immobilised Bacteria

The batches were analysed microbiologically and biochemically at four time points. The analyses showed that the lignin + PVP carrier significantly influenced the total bacterial count and the dehydrogenase activity. The multiple regression analysis proved that the second-degree model was the best model for both parameters (the count and microbial activity). None of the higher-order models showed a better fit, and the $R^2$ coefficient of determination assumed the highest values in the quadratic model.

The analysis of the average results noted in individual experimental variants shows that the highest count of eubacteria was in the WAF + SS + Lignin variant. It can be seen in Figure 8 and in the fluorescence microscope image (see Figure 9).

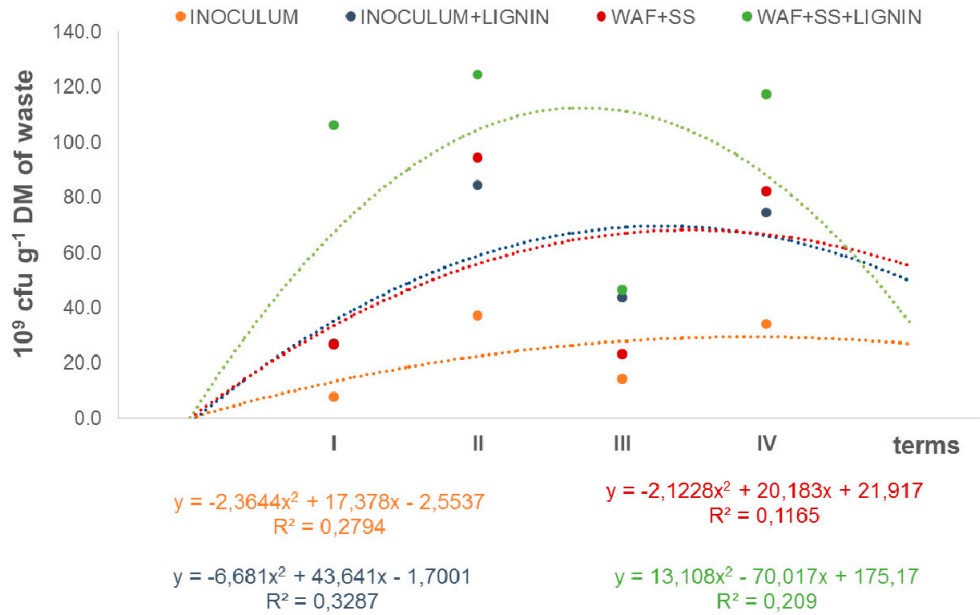

**Figure 8.** Variation in the total count of heterotrophic eubacteria in the samples subjected to anaerobic digestion.

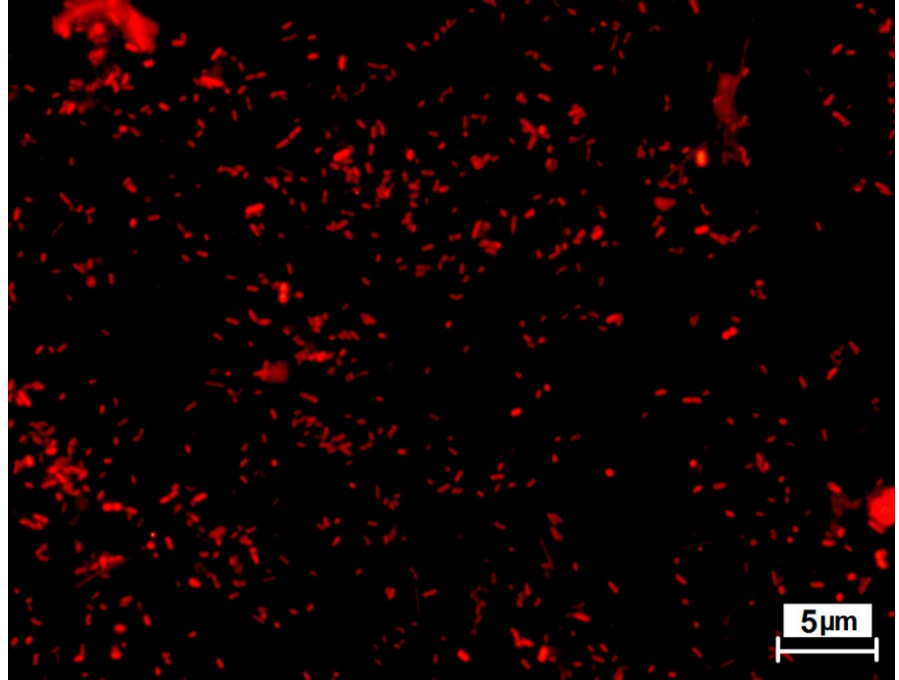

**Figure 9.** Specific identification of whole fixed bacterial cells with fluorescent oligonucleotide probes (FISH) in the WAF + SS + Lignin sample.

The addition of PVP-modified lignin to the WAF + SS sample intensified bacterial proliferation and reached the highest value of $124 \times 10^9$ cfu $g^{-1}$ DM at the second time point. By comparison, at the same time point the bacterial count in the sample without the carrier was $94 \times 10^9$ cfu $g^{-1}$ DM. The average results noted during the experiment indicate that the bacterial count increased by 46%. According to scientific reports, it is possible to achieve a high increase in cell concentration in immobilised biomass [4].

There were very good results of bacterial cell immobilisation on lignin in our experiment. There have been few reports describing the use of lignin as a carrier or carrier component in

biotechnological processes [14,33]. The reports indicated that lignin was characterised by very good sorption and it could be successfully used as a carrier for the immobilisation of bacteria and enzymes.

Recently Gong et al. (2017) [14] proved that $\alpha$-amylase could be immobilised by means of lignin from bamboo shoot shells (BSS). The study showed that lignin was an ideal $\alpha$-amylase activator as it increased the enzyme activity more than two-fold at a concentration of 5 mg/mL. The activating effects of BSS lignin can be partially ascribed to its coarse surface, where numerous pits provide an optimal location for stable contact between the support and enzyme. However, as a support for $\alpha$-amylase, BSS lignin is not devoid of certain limitations. Most among them, the lower stability of incubation at 60 °C and over 30 min [14]. Ciesielczyk et al. (2014) [33] also used kraft lignin to develop a hybrid system. They combined lignin and the synthetic inorganic support MgO · SiO$_2$. The material underwent detailed physicochemical analysis, which showed its valuable properties and qualified it for being used as an effective adsorbent in environmental protection. The assessment of the hybrid system was not only affected by the favourable results of its dispersive and morphological characteristics, but also its electrokinetic test results and thermal properties, confirming its stability over a wide range of temperatures and pH values. This information is important also in the aspect of the choice of lignin as a microbiological carrier in anaerobic digestion.

An SEM image of the WAF + SS + Lignin sample (with the largest number of microorganisms) was additionally taken to prove the colonisation of the lignin surface (Figure 10). The image shows clusters of microorganisms, mostly rod-shaped bacteria. It is most likely that the bond between cells and the lignin + PVP carrier has covalent nature. The mechanism of formation of this bond was presented in the study by Lalov et al. (2001) [6]. The researchers found that the methanogenic microorganisms bound to the synthetic polymer via hydroxymethyl groups from the support and probably via amino groups from the methanogenic cells [6]. It is known that the cell walls of methanogens often contain proteins and other organic compounds with amino groups, which may take part in that reaction Covalent cell immobilisation on the carrier is more durable and beneficial. Covalent cell immobilisation on the carrier is more durable and beneficial.

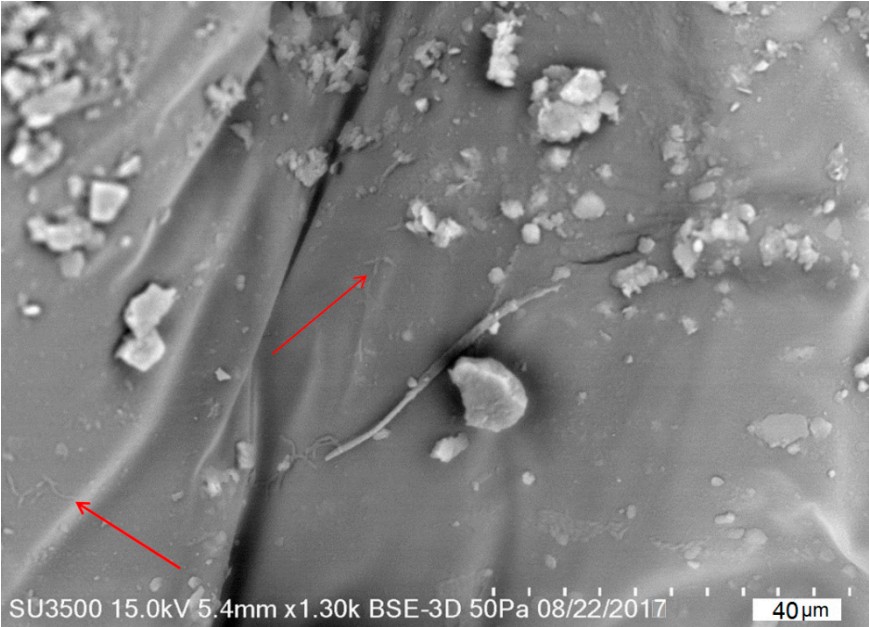

**Figure 10.** SEM image of the lignin surface colonised in the WAF + SS + Lignin sample.

In order to increase cell adhesion to the carrier surface PVP was used in this study. This polymer was also used in the study by Singleton et al. (2002) [19]. PVP can bind toxic compounds in the medium and has a high water-binding capacity. According to recent reports, the addition of specific micronutrients to the microbial support increases the microbial film capacity to attach and grow on the

support surface [41]. Experiments proved that when a carrier is prepared in this way, it may effectively reduce inhibitor substances in the anaerobic digestion of wastewater.

The digestate produced in the AD process was also analysed biochemically to measure the level of dehydrogenases (intracellular enzymes, DHA), which are commonly recognised as bacterial activity indicators [42].

As Figure 11 shows, the DHA in the samples with lignin was noticeably greater than in those without the carrier. The average results of analyses made during the experiment show that the dehydrogenase activity in the WAF + SS + Lignin system increased by as much as 43%. It is another fact proving that lignin or PVP-coated lignin is an excellent microbial carrier. Cells not only proliferate more intensely on lignin, but they are also more active. In consequence, the process of substrate biodegradation is intensified. The results of the biochemical analyses presented in Diagram 7 show that in most of the samples the DHA was increasing gradually until the third time point—the 8th day. It is most likely that the increase was caused by the availability of easily decomposable organic matter.

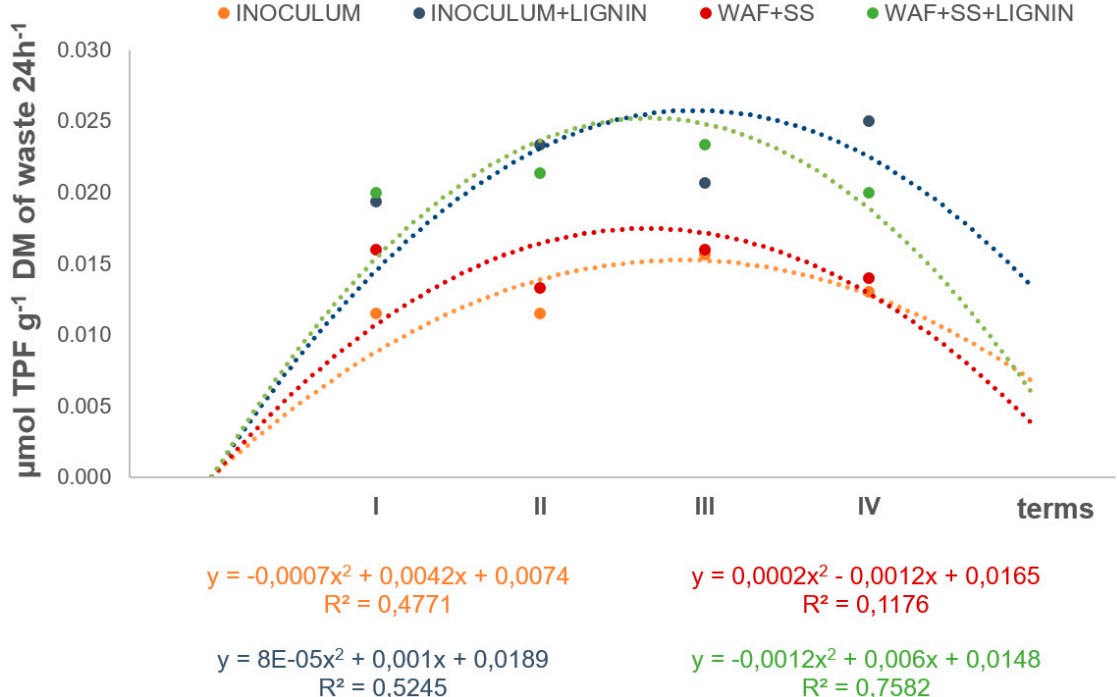

**Figure 11.** Variation in the dehydrogenase activity (DHA) in the samples subjected to anaerobic digestion.

### 3.3.3. Biogas Production Performance

During the experiment the biogas/methane production proceeded successively every day (according to the shape of the lines in the diagrams in Figure 12a,b) until the 17th day (in the WAF + SS sample) and until the 20th day (in the other variants). The production process was stable, without inhibition, as the shape of the curves shows.

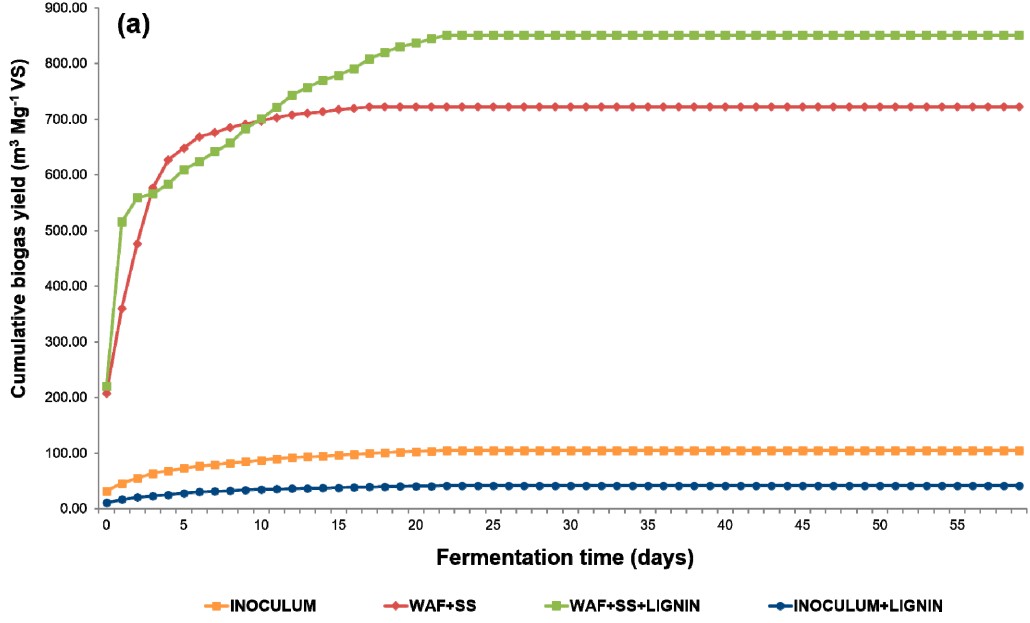

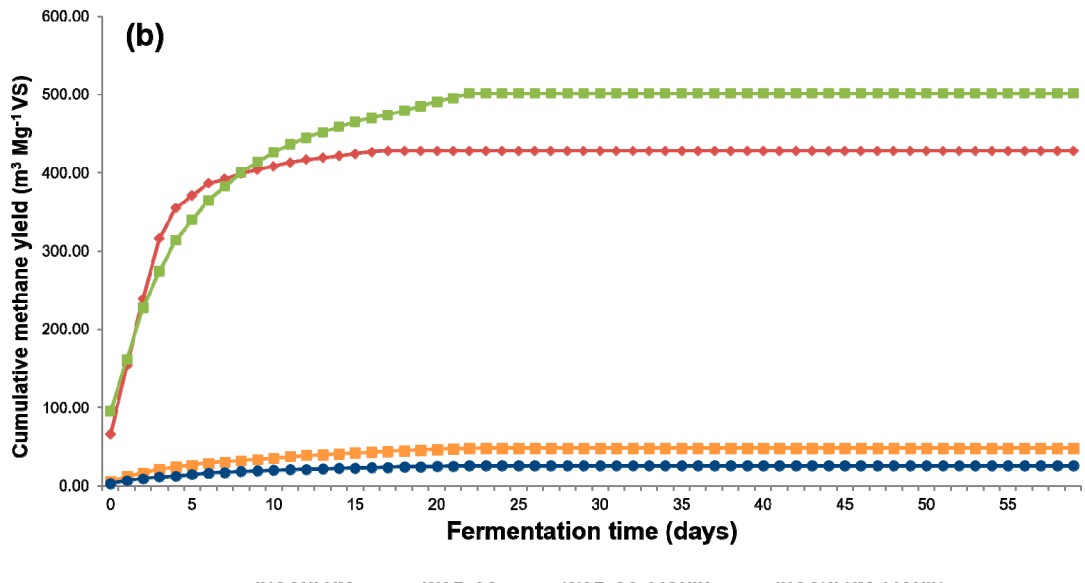

**Figure 12.** Cumulative biogas (**a**) and methane (**b**) production curves from VS of the WAF+SS WAF + SS + Lignin, Inoculum, Inoculum + Lignin samples.

The biogas yield noted in this experiment corresponds to the abovementioned results of microbiological and biochemical analyses. The variants with the lignin + PVP carrier produced more biogas/methane. Diagrams Figure 13a,b show the results of cumulated biogas and methane in terms of fresh matter (FM) and organic matter—b (VS). The WAF + SS combination produced 722 m$^3$ Mg$^{-1}$ VS of biogas in the AD process, including as much as 58% of methane, i.e., 428 m$^3$ Mg$^{-1}$ VS. The same batch with the lignin + PVP carrier released a larger amount of biogas (850 m$^3$ Mg$^{-1}$ VS) through anaerobic biodegradation. The amount of methane (503 m$^3$ Mg$^{-1}$ VS) increased by almost 18%. The amount of biogas (including methane) produced by the WAF + SS sample was consistent with the results noted by the author of this article in her earlier studies [17]. The results of the AD process efficiency in the samples with and without the carrier were comparable with the results of experiments given in reference publications, including co-digestion combinations between sewage sludge and other types of organic waste [38].

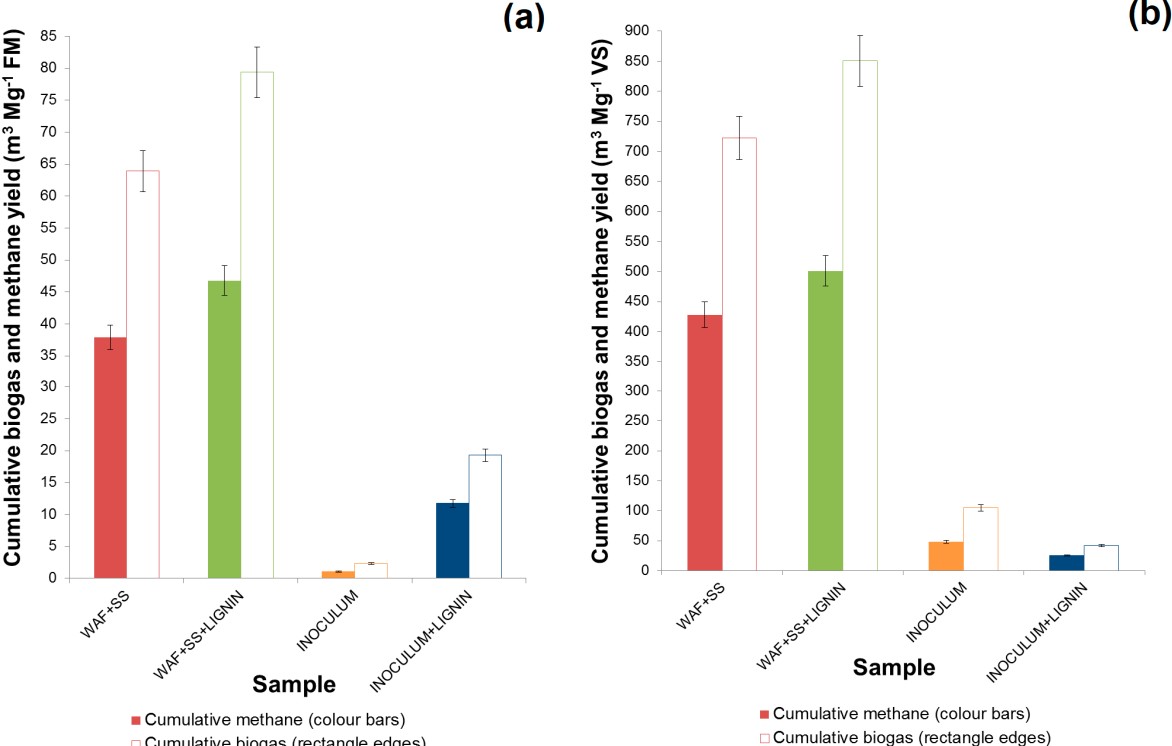

**Figure 13.** Cumulative biogas and methane yield from the fresh matter (FM) (**a**) and volatile solids (VS) (**b**) of the WAF + SS, WAF + SS + Lignin, Inoculum, Inoculum + Lignin samples.

Lalov et al. (2001) [4] reported improvement in biogas production resulting from cell immobilisation. The researchers used a granulated polymeric support [poly(acrylonitrile-acrylamide)] and obtained 22% more methane from immobilised biomass than from free biomass. Borja et al. (1994) [43] tested zeolite as a carrier in an AD reactor and observed a 59% increase in specific bacterial growth and about a 20% increase in the methane yield. After using lignins as a carrier the increase in the amount of biogas/methane produced in this experiment was comparable with other researchers' results.

The lignin additive not only increased the count and activity of microorganisms and the biogas production efficiency, but it also adsorbed heavy metals. According to reference publications, researchers successfully used kraft lignin in studies on the biosorption of different heavy metals contained in wastewater [12]. It is possible that in our experiment lignin also adsorbed some heavy metals or other toxic compounds from the SS co-substrate. The use of lignin may solve various problems occurring in the AD process based on materials with unfavourable composition, such as different types of waste products. Further research should be oriented at this target. The author of this study will conduct further research to verify whether lignin could be not only a cell carrier, but also an effective heavy metal adsorbent in anaerobic digestion of sewage sludge.

Both confectionery waste (including waste wafers) and sewage sludge are of equal significance in anaerobic co-digestion. Both materials meet all the conditions of successful anaerobic co-digestion with synergistic effects [17,18]. The factors attributed to the synergistic effects are inherently associated with the properties and composition of co-substrates [40]. As was mentioned before, sewage sludge is an excellent factor buffering the system, whereas confectionery waste effectively balances the C/N ratio and the content of macro- and microelements. Thus, the anaerobic co-digestion of both substrates is economically feasible and practical. So far there have been numerous publications on the use of sewage sludge as a substrate in the AD process and the number of these reports is systematically growing [29,44]. However, there have been few reports on the use of confectionery waste despite its high potential [16]. Research on the efficient use of both substrates for methane production and

the application of natural, available microbial carriers should be continued for environmental and economic reasons. It is also justified and necessary to continue research on the application of carriers in anaerobic digestion. The positive results of this study show that these carriers have a high potential for successful application in industrial production.

## 4. Conclusions

The research results show that kraft lignin grafted with PVP had a positive effect on the anaerobic co-digestion of waste wafers and sewage sludge. The porous microstructure of PVP-coated material promoted cell adhesion and increased cell proliferation and activity. The cell count increased by 46% in the WAF + SS variant with the carrier, whereas the enzyme activity increased by 43%. It was a very good result and it was comparable with the results given in reference publications for other carriers tested in anaerobic digestion. The potential use of lignin for biogas production in the AD process is also determined by its excellent thermal stability and the fact that it is not degraded by hydrolytic enzymes.

In consequence of improvement in the microbiological and biochemical parameters, there was a natural increase in the amount of biogas/methane produced. The WAF + SS co-substrate system generated 722 $m^3$ of $Mg^{-1}$ VS of biogas, including 428 $m^3$ of $Mg^{-1}$ VS of methane, whereas the immobilised WAF + SS + Lignin sample generated 846 $m^3$ $Mg^{-1}$ VS of biogas, including 502 $m^3$ $Mg^{-1}$ VS of methane. A nearly 18% increase in the volume of biogas/methane produced in the AD process was comparable with the results of tests on other carriers given in reference publications.

To sum up, it is noteworthy that food waste from the confectionery industry, combined with sewage sludge, has a great practical potential. It is an interesting solution which can be used to produce renewable energy. It provides new substrates to agricultural biogas plants and is remarkable in the aspect of environmental protection. There are similar observations about the use of lignin waste. The proposed method enables the disposal of lignin production waste in anaerobic digestion and provides realistic benefits as it improves the process efficiency. The potential of lignin in anaerobic digestion processes cannot be overestimated. It can be used not only as a microbiological carrier, but also as an adsorbent of heavy metals in the sewage sludge.

**Author Contributions:** A.A.P. conceived and designed the experiments, performed selected analyses and wrote the paper. A.W.-M. performed microbial and biochemical analyses. K.P. performed batch digestion test.

**Funding:** This research was funded by the company Biolab-Energy A&P, Poznań, Poland.

**Acknowledgments:** The English language of this article has been improved by Richard Ashcroft, a freelance scientific copy editor.

**Conflicts of Interest:** The authors declare no conflicts of interest.

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
