# Peer review of "Kraft Lignin Grafted with Polyvinylpyrrolidone as a Novel Microbial Carrier in Biogas Production"

_energies, doi:10.3390/en11123246_

Round 1
Reviewer 1 Report
Submission to be reviewed: Kraft Lignin Grafted with PVP as a Novel Microbial Carrier in the Biogas Production Process
This paper deals with use of kraft lignin grafted with PVP.
I appreciate the work of the authors; however, there are still issues that needs to be addressed before potential acceptance of this contribution.
The comments to be resolved are below:
Abstract needs to be improved. Abstract should give the aim of the contribution, basic methodology (objects of the research, and research and analytical methods), principal findings (give specific data and their statistical significance), and the main conclusions. In addition, authors need to emphasize new and important aspects of their study. Furthermore, there are “authors” of this contribution, however in the abstract is term “author”.
The author / authors issue is throughout the manuscript. How should reader understand this issue?
In addition, this is the reason, why section “Author contributions” is important.
The formatting of the paper is somehow confusing. However, this is rather technical issue.
Why authors use “AcoD” abbreviation for anaerobic co-digestion? As this is not a common abbreviation, therefore I recommend revising this.
Generally, the English is not sufficient and paper must be proofread by native speaker.
Do not use same words in keywords and title.
Paper is quite long and in some areas repetitive. I would recommend authors to make their contribution more concise.
There is a number of rather poor statements, i.e.: “The immobilisation process should be simple and harmless”. This is more of a textbook style than research paper.
The Introduction is too much combined with authors intentions, such as “the authors of this study decided to test lignin”.
Authors need to revise the Introduction section and significantly improve it.
In Materials and Methods section, the origin/meaning of “Stuffed wafers (WAF)” in the paper is unclear.
Statements such as “waste materials used in the research” is useless for the reader, unless it giver further explanation.
Why authors decided to use “m3 Mg-1” for description of biogas yield?
Chapter 2.3. needs to be improved.
Microbial analysis of digestate – this chapter has very unclear connection to the rest of the paper. It is essential, that scientific paper has a clear linkage throughout the paper – what you explain as a problem in Introduction, you show how you do it in M+M and afterwards describe in R+D. However, this linkage is currently missing.
How can authors describe results of “Thermal analysis”, when there was not sufficient background for it in Introduction or M+M? And number of similar issues in the paper.
Specific comments:
The Tables and Figures in “Energies template” can be put directly in the text.
When describing multifunctional portable gas analyser, provide the full information. I.e.: Biogas composition was measured using a GA5000 multifunctional portable gas analyser (Geotech, Leamington Spa, UK), which is adapted to measurements of CH4, CO2, O2, H2, and H2S with the following measurement accuracy of: CH4 (0–70 vol % ± 0.5%), CH4 (70–100 vol % ± 1.5%), CO2 (0–60 vol % ± 0.5%), CO2 (60–100 vol % ± 1.5%), O2 (0–25 vol % ± 1.0%), H2S (0–5000 ppm ± 2.0%), and H2S (0–10,000 ppm ± 5.0%). See description of Materials and Methods in the following paper which also used Geotech GA5000 https://www.mdpi.com/1996-1073/11/7/1794/htm
Same for other used devices and analysers.
In conclusion, even though I can clearly see that the authors did significant amount of work, the paper in its current state is very confusing and not ready to be published. However, if the authors will significantly improve the paper and its structure, it can be considered further.
With best regards,
The Reviewer
Author Response
Poznan, 12-11-2018
Dear Reviewer,
To begin with, I wish to thank you for your time, for reading and evaluating the paper written by myself and my colleagues, and for your valuable suggestions.
1. Indeed, the word “author” in the Summary was meant to be “authors”. This has been corrected as above and checked in the whole text.
2. We have corrected the Abstract in detail, as suggested.
3. The “AcoD” abbreviation for anaerobic co-digestion was used as suggested in the literature and it means the same as “AD”. However, it has been deleted, as you suggested.
4. The whole text has been corrected by a native speaker (see Acknowledgements).
5. The key words have been selected so as to avoid repetition of the words in the title.
6. I have deleted a few fragments which seemed to be repetitions of ones in the Introduction and Results and discussion.
7. These awkward expressions have been reworded. Some of them have also been detected and corrected by the native speaker.
8. New information has been added in the Introduction and some other has been deleted from it, as suggested. I do hope my efforts have improved it and will be appreciated.
9. The fact that we used waste materials is very important because we aimed to utilize food waste for biogas production – this has been widely practiced for some time already. In particular, we wish to arouse interest in the utilization of food waste from the confectionery industry (Pilarska et al., 2018). This is important in the aspect of environmental protection and is also of great concern to readers who understand the authors’ intentions – that is, to the organic-waste biogas production specialists. General information on the waste source can be provided but I am not always permitted to fully identify the provider of the waste material for obvious reasons.
10. This is a unit I always use – so do other researchers in other countries. The unit expresses the generated amount of biogas/methane. In most journals, the amount is referred to as “performance” or “efficiency”.
11. I regret to have to observe that I am not familiar with any other method to describe such analyses or analyzers. The descriptions, as they are, extend to many pages and are as comprehensive as possible. More details can be obtained from catalogues by those particularly interested. According to the literature and my experience (Pilarska et al., Powder Technology, 2013; Klapiszewski et al….in References), such descriptions are used as standard ones for this kind of analyses and they look just as they do in this case.
When introducing a microbiological carrier into a system (specifically, into one with heating), thermal analysis has to be carried out to make sure the material will remain stable and will not decompose in the given conditions. Other analyses which were described in the paper are also important. In the Introduction, it has been explicitly stated what kind of properties potential microbiological carriers ought to have. The lignin materials were analyzed in this research work just to verify they do have such properties.
The information has been added also in the Introduction, as suggested. Moreover, the Introduction has been condensed to make it less redundant and more compatible. Thank you for your advice.
12. Thank you very much for indicating the link. I have used the website information.
Best regards,
Agnieszka Pilarska

Reviewer 2 Report
Authors demonstrated the kraft lignin grafted with the efficiency of polyvinylpyrrolidone (PVP) as a novel microbial carrier anaerobic co-digestion (AcoD). The author further characterized the morphological structure, dispersive and adsorptive properties of lignin and combination of lignin with PVP. The research concluded that lignin is effective microbial carrier material for biogas productivity enhancement in anaerobic digestion (AD). However, I have the following comments
1. Title of the manuscript should avoid the abbreviation of PVP replacing full form. Furthermore, “process” word is not necessary.
2. The graphical abstract is not clear at all and hard to follow. I would encourage the author to improve the graphical abstract.
3. Line 56 “available reference publications” should replace by recent state-of-the-art
4. The carrier materials have a significant role to build and retain the biofilm of selected microbial dynamics. Principally, microbial and material interaction leads to form the biofilm and thereby degrade the substrate. Thus, selection of high-efficiency biofilm carrier plays an essential role for the enrichment of high-density methanogen, and these tactics may be available to prevent the biomass being washed out in the effluent. Previously reported studies have shown the retention of microorganism in AD reactors by biofilm carrier can potentially increase their productivity by increasing the amount of methanogens and or other mix culture. Authors completely ignore these facts and have not been describing the material microbes-interaction, possible mechanism to enhance the production and some of the ideal condition for biofilm formation. I would encourage to elaborate in the introduction part, perhaps additional reference might be supportive to describe the microbes-material interaction and biofilm formation i) An overview of cathode materials for microbial electrosynthesis of chemicals from carbon dioxide. Green Chemistry, 19(24), 5748-5760. (2) On the edge of research and technological application: a critical review of electromethanogenesis. International journal of molecular sciences, 18(4), 874.¨
5. The BET surface area of Lignin is 1.9 m2/g, whereas the BET surface area of lignin+PVP was 2.5 m2/g. What was the reason? Why was the surface area increased?
6. What was the activity author trying summarized In line 699?
Author Response
Poznan, 12-11-2018
Dear Reviewer,
Thank you very much for your valuable suggestions and for indicating the source of more information. It will certainly be of use in my future work.
The abbreviation PVP in the title has been replaced with the full name and the word “process” has been deleted.
The graphical abstract has been improved and is more legible now.
I have used the expression suggested by you – thank you very much for your advice.
Based on the references suggested by you, I have added some valuable information on the microbes-material interaction and biofilm formation (marked in color) in the Introduction.
These results were obtained in our analyses. According to my observations regarding various materials (including MgO), an increase in surface area may be observed with an increase in the porosity of materials and this happened here; also the presence of certain irregularities and increase in particle size may have an effect on the increase in the BET surface area.
A short explanation has been added.
Here, I meant an increase in enzymatic activity (dehydrogenase activity was studied).
Best regards,
Agnieszka Pilarska

Reviewer 3 Report
This is surely an interesting and original work. It has the potential to be published in Energies. I have only a couple of comments that the authors should implement into the revised manuscript prior to publication.
1) Conclusions - The practical impact of the results obtained in this work should be better highlighted.
2) Referecens - References are too extensive. Are they really all necessary?
Author Response
Poznan, 12-11-2018
Dear Reviewer,
Thank you very much for your very good opinion on my paper and for your valuable suggestions.
- A mention on the practical aspects of my method has been added in the Conclusions;
- The literature has been reduced by deleting 12 items. Although they all seemed indispensable at first, the previous final number of References was too high, indeed.
- The whole text as been verified and corrected by a native speaker (see Acknowledgements).
Best regards,
Agnieszka Pilarska

Round 2
Reviewer 1 Report
In this paper, even though authors significantly improved their work, there are still a few issues.
I.e.: Line 345 – “According to some authors” – you need to clearly write what authors. Statement such as “according to some authors” without referring to real papers does not belong to the research paper.
The authors should consider enlarging the “discussion” in the paper.
The Tables and Figures in “Energies template” should be put directly in the text. As in the current version I am not able to review none of the Tables or Figures.
I appreciate the graphical abstract.
Author Response
Poznan, 16-11-2018
Dear Reviewer,
To begin with, I wish to thank you for your time, for reading and re-teview our the paper.
Thank you also for your valuable suggestions.
- The whole text has been verified and corrected again - by a native speaker;
- I have made changes in yellow;
- I enlarging discussion in the text.
Best regards,
Agnieszka Pilarska

Reviewer 2 Report
This paper is strongly recommended for publication.
Author Response
Poznan, 16-11-2018
Dear Reviewer,
Thank you very much for your very good opinion on my paper and recommendation.
The whole text has been verified and corrected again - by a native speaker.
Best regards,
Agnieszka Pilarska

Reviewer 3 Report
The authors have addressed my comments in a satisfactory manner. Overall, the manuscript has been improved after revision. Therefore, it can be accepted for publication in Energies.
Author Response
Poznan, 16-11-2018
Dear Reviewer,
Thank you very much for your very good opinion on my paper and recommendation.
The whole text as been verified and corrected again - by a native speaker.
Best regards,
Agnieszka Pilarska

Round 3
Reviewer 1 Report
After reviewing this paper for the third time, I believe authors did a significant amount of work in order to improve presentation of their research. The research itself was interesting from the beginning, but for the future, authors should keep in mind to provide precise manuscript which would be free of typos from the beginning.